# “What Do You Need? What Are You Experiencing?” Relationship Building and Power Dynamics in Participatory Research Projects: Critical Self-Reflections of Researchers

**DOI:** 10.3390/ijerph19159336

**Published:** 2022-07-30

**Authors:** Doris Arnold, Andrea Glässel, Tabea Böttger, Navina Sarma, Andreas Bethmann, Petra Narimani

**Affiliations:** 1Department of Social Work and Health Care, Ludwigshafen University of Business and Society, 67059 Ludwigshafen, Germany; 2Institute of Biomedical Ethics and History of Medicine, University of Zurich, 8006 Zurich, Switzerland; andrea.glaessel@ibme.uzh.ch; 3Institute of Public Health (IPH), Zurich University of Applied Sciences (ZHAW), 8400 Winterthur, Switzerland; 4Institute of Health Science, Faculty of Medicine, University of Lübeck, 23562 Lübeck, Germany; tabea.boettger@uni-luebeck.de; 5Department of Infectious Disease Epidemiology, Robert Koch Institute, 13353 Berlin, Germany; sarman@rki.de; 6Centre for International Health Protection (ZIG), Robert Koch Institute, 13353 Berlin, Germany; bethmanna@rki.de; 7Protestant University of Applied Sciences Berlin (EHB), 14167 Berlin, Germany; petra.narimani@aol.de

**Keywords:** participatory research, participation, health, power, reflection, research relationships, understanding of roles in research, error culture, DIPEx

## Abstract

Participatory approaches create opportunities for cooperation, building relationships, gaining knowledge, rethinking, and eventually changing power structures. From an international perspective, the article looks at the historical development of different participatory approaches in which building relationships and managing the balance of power between persons engaged in participatory research are central. The authors present and critically reflect on four research projects to show how they understood and implemented participatory research in different ways and what they have learned from their respective experiences. The “PaSuMi” project worked in the context of addiction prevention with migrants and provides a glimpse into different contexts of participatory research. The initiator of the study “Back into life—with a power wheelchair” works with post-stroke individuals who use the assistive device in community mobility and reflects on the shifting and intertwining roles of participants. In the research project “Workshops for implementation of expanded community nursing”, new professional roles for nurses in community nursing were developed; here limitations to participation and ways to deal with them are illustrated. Finally, the “DIPEx” project deals with challenges of enabling participation of persons with multiple sclerosis via narrative interviews on the experience of health and illness. All examples underline the necessity of a permanent reflection on relationships and power dynamics in participatory research processes.

## 1. Introduction

The concept of participation in practice and research is ambiguous and multifaceted and must therefore largely adapt to the respective living environments with their different socializations, experiences, emergencies, and needs. Participatory approaches are, therefore, not only dependent on different conditions, but they also have to be designed differently at different levels [1] (pp. 445–453) [2]. Involvement and participation in social decision-making processes always have been and still are a matter of course in many nations and cultures. As an example, some Latin American countries such as Ecuador or Chile—and recently Colombia—are mentioned here, in which quite a few of the numerous nations and ethnic groups want to maintain their ways of life based on very different forms of participation. At the same time, they are striving for a plurinational superordinate state whose tasks they would like to co-determine.

Participatory research approaches are rooted in social movements that stand up for a democratic and inclusive society. Among others, these include Participatory Rural Appraisal (CHAMBERS), Emancipatory Research Approaches (FREIRE), Action Research in Organizational Development (LEWIN), Action Research in Education (KEMMIS), Human Inquiry and Cooperative Inquiry (REASON), Appreciative Inquiry (COOPERRIDER), Community-Based Participatory Research (WALLERSTEIN), Action Science (ARGYRIS), Constructivist Research (LINCOLN), Feminist Research (LATHER), Empowerment Evaluation (FETTERMAN), and Democratic Dialogue (GUSTAVSEN) [3].

Participatory research in Germany draws on many different sources and traditions [4]. Its common denominator might be that the people and/or groups involved shape the research process as independent and equal actors. Their concerns, inequalities, and lives are at the core of the joint research processes as well as the intention to bring about change. The design of the relationship of all actors involved is of central importance for participatory research. At this point, different interests, roles, and power dynamics are inevitably at stake, as well as structural obstacles within and outside the research community. Thus, the issue of relationships in the research process represents a challenge as well as an opportunity to critically reflect on, improve and enhance participatory research.

Building on North American and South American approaches in the mid-20th century, “action research” flourished in the wake of the student movement of the late 1960s in Germany, Austria, and Switzerland. In 1993, Altrichter and Gstettner critically examined the booming publications on action research in the German-speaking world between 1972 and 1982 [5,6]. According to findings from a survey in 1990 action research had nearly disappeared from the German social science debate at that time, mostly due to its decreased attractiveness [5]. It came back as participatory research in the middle of the first decennial in the new millennium. Recently, we even saw an increased financial support of participatory research projects in Germany. Nevertheless, the definition of participatory research often seems to be unclear, both for researchers and for funders [4]. Currently, there are increasing discussions about suitable criteria and forms of funding for participatory research projects, which could lead to clearer regulations for funding participatory research in the near future [7,8]. On the other hand, Green and Johns [9] warned against “token” participation and pointed to recent pressure from ministries to make projects participatory.

Looking at the issues compiled by Altrichter and Gstettner on alleged and actual weaknesses of the action research approach today (in 2022), we find that many are still relevant and unanswered in Germany and worldwide. This article will focus on the issue of “relations between researchers and researched” [5] (p. 69) raised by Altrichter and Gstettner. In the English discourse on participatory health research, challenges connected with power and positionality of researchers in working with co-researchers are addressed in different contexts. Drawing on data from workshops with an international sample of participatory health researchers, Egid et al. [10] shed light on these issues and the need for reflexivity as a means for researchers to address power inequities in participatory research practice. Smith et al. [11] reflected on different positionalities of class, gender, race, sexual orientation and the status of insider or outsider of communities, using episodes taken from the experiences of novice participatory action researchers. In the context of CBPR, Muhammad et al. [12] discussed issues of identity, intersectional positionality and power dynamics by means of autoethnographic self- reflections of experienced researchers in working with partners from marginalized communities. In a slightly different vein, Jagosh et al. [13] showed the crucial importance of trust, power-sharing and co-governance for CBPR projects to be successful, employing a qualitative analysis of interviews with community members and researchers.

In this article, we discuss individual experiences in four research projects in the German context. We look at relationships and power dynamics between persons engaged in a participatory research process. By reflecting challenges and by showing how we understood and reacted to them, on one hand special features and the high value of participatory health research are emphasized. On the other hand, limitations to these endeavors are made transparent.

## 2. The Importance of Research Relationship and Power Dynamics from a Methodological Perspective

Facilitating participation in research processes has implications for the structure of relationships in the context of existing power dynamics among persons engaged in a participatory research process. With this essential characteristic the participatory research approach [14] differs from other approaches in social and health research. In this article, we refer mostly to the approach of Participatory Health Research (PHR).

The international Collaboration for Participatory Health Research (ICPHR) states the goal of PHR and the demands on participation in the research relationship within PHR as follows:

“The goal of PHR is to maximize the participation of those whose life or work is the subject of the research in all stages of the research process, including the formulation of the research question and goal, the development of a research design, the selection of appropriate methods for data collection and analysis, the implementation of the research, the interpretation of the results, and the dissemination of the findings. (…)” [15] (p. 6).

Participation refers to the co-operation in decision-making and research activities of those persons who are directly affected by the problems that are the subject of the respective research projects. Participatory Health Research understands participation as co-learning, co-operation or partnership in decision-making within research processes [16]. This requirement reaches beyond mere participation in the sense of consultation and involvement in data collection or analysis, as is also included, for example, in feminist, ethnographic, and other qualitative research designs (see, for example, [17] pp. 142–146). In participatory research and qualitative research designs, initiators of research projects should establish relationships based on trust with persons directly addressed by the research and with other stakeholders involved. In addition, common features of participatory research and other qualitative research designs, such as ethnography, might include creating spaces for hearing voices of persons in marginalized positions [18] (p. 435) and including their perspectives.

Thus, participation in research always addresses the relationship and power dynamics between researchers and the persons who are at the center of research processes, with and through whom knowledge on specific issues is generated [9,10,11,12,19]. At the same time, participatory research has the potential to enable the persons involved to bring about change within their communities and/or their individual situations through their co-operation in the research [20]. Epistemologically, the question here is how and in which social processes knowledge can emerge. Participation in research therefore has implications for epistemological aspects as well as for the design of the collaboration between researchers and other persons engaged in participatory research projects.

Issues concerning the design of research relationships and the epistemological aspects mentioned pertain to approaches of participatory research as well as to other qualitative research approaches and differentiate them from the prevailing positivist/post-positivist concept of science in quantitative research [21]. Thus, in quantitative research, the research relationship is supposed to be distant and characterized by objectivity, whereas in participatory research, as mentioned above, this issue is understood in a fundamentally different way due to the need to build a collaborative and trusting research relationship. For this reason, among others, participatory research or action research approaches had to grapple with criticism from established social research communities regarding the lack of “objectivity” and the scientific credibility of its results since their inception [5].

If participatory research is closely oriented towards the persons about whom it generates knowledge, and researchers work closely with these persons, researchers are at the same time consciously taking sides with persons who often belong to groups that are marginalized or disadvantaged in society. With reference to the history of action research in the German-speaking countries, one of the controversial issues discussed is the role of “admitted partiality” advocated by some action researchers [5] (p. 71). Hereby, the positivist/post-positivist claim to objectivity of conventional quantitative research was deliberately questioned in favor of a critical, politically motivated positioning of researchers on the side of those affected by the exercise of power and discrimination. Thus, in the early 1980s, Maria Mies assigned “partiality” a central place for feminist research in her “Methodological Postulates for Women’s Studies” [22]. Mies’s conception of feminist research included a clear commitment to politically motivated action research (with reference to Paolo Freire, among others) and understood “partiality” in favor of discrimination against women as a critique of the claim to absence of bias in the prevailing positivist/post-positivist understanding of science.

However, these epistemological and methodological implications of participation not only apply to participatory research, but also to other research styles of qualitative research, which, according to Yvonna Lincoln and colleagues, are oriented towards alternative inquiry paradigms [21,23]. One of the common characteristics of qualitative research approaches is the concept of research as a “situated activity” [24] that locates researchers and research activities in the social worlds of and thus in social interactions with the persons at the focus of research processes [23] (p. 10).

With reference to the German-language discourse, Hella Von Unger [20] mentioned three distinguishing features between participatory research and other interpretative, qualitative research styles:(a)Co-operation of co-researchers in the research process on an equal basis.(b)Initiation of “… learning process[es] that enable individual and collective strengthening and development processes…” among the persons involved in the research [20] (p. 162; translation by the authors).(c)The “dual objective” of participatory research: firstly, to generate knowledge that contributes to a better understanding of the respective underlying problem, and, secondly, to bring about processes of change (ibid.).

According to Lincoln and colleagues [21], key differences between positivist/postpositivist research and qualitative research are, among others, tied up to the following questions about control over and participation in the research process:

“Who initiates? Who determines salient questions? Who determines what constitutes findings? Who determines how data will be collected? Who determines in what form the findings will be made public, if at all? Who determines what representations will be made of participants in the research?” [21] (p. 134).

This is particularly evident when compared with experimental studies (e.g., randomized controlled trials; [25]), which have played an increasingly powerful role in decision-making pertaining to policies in health care since the beginning of this century in the pursuit of evidence-based health care. Key characteristics of these studies are that control and decision-making power during the entire research process rest solely with the researchers. Any relinquishment of control over any of the aspects addressed in terms of participation in research would run the risk of creating bias, and thus would have a major detrimental impact on the quality of the study results (ibid.).

The questions mentioned above are answered in a decidedly different way in participatory research and at the same time address the significance of power in the research relationship between researchers and persons in the focus of the research. The importance of participation in terms of influence and co-design in the research process will be exemplified in the following examples from our research practice.

## 3. Four Examples of Designs of Research Relationships and Participation

### 3.1. Approach to Critical Self-Reflection on Research Experiences

The following examples are based on discussions and critical reflections on participation and participatory research that took place during an ICPHR Training in Participatory Health Research program in 2018 at the Catholic University of Applied Sciences, Berlin. We present four of the projects that were accompanied by all participants during the training in the form of collegial consultation. All projects were independent of each other in terms of time, content and aims. We chose an approach of critical self-reflection that in a similar way has been used by other researchers for sharing experiences and enhancing knowledge about the practice of participatory research [11,12]. First, we briefly introduce the aims, design and context of each project and describe their role as researchers or initiators of the projects. In a second step, we critically reflect our personal experiences as well as challenges during the research process that relate to issues of power dynamics and relationships. Finally, we point out aspects that facilitated or limited participation in research processes.

### 3.2. “Nothing about Us, without Us”—Participation in Addiction Prevention among Migrants—Reflections on Power Sharing and Roles within the PaSuMi Project by the Author Navina Sarma

#### 3.2.1. Introduction

The PaSuMi (Participation, Addiction Prevention and Migration) project, duration 2017–2019, was a nationwide participatory model project of the German AIDS Service Organization (DAH) in the context of addiction prevention among migrants (https://pasumi.info/, (accessed on 26 July 2022)). Persons from and around eight addiction support and primary addiction prevention facilities in five German cities were involved. The common goal was to improve access to the help system with and for migrants. Clients, activists and community partners developed model approaches in a co-operative manner as research partners alongside employees of addiction support facilities (hereinafter referred to as practitioners) and academic researchers of the DAH [26].

In the first project phase, the project coordinators of the eight facilities formed local project teams. While one team consisted exclusively of activists from a self-organized initiative of Russian-speaking persons who use drugs (BerLUN) and persons from their community, other teams included staff (e.g., social workers, counselors, language mediators), (former) clients (e.g., from counseling or consumption rooms) and persons from specific communities (e.g., refugees, sex workers). Researcher partners called themselves peers or community partners (hereafter community partners). The size and composition of the project teams varied and was dynamic depending on the upcoming activities.

In the PaSuMi project, persons who previously acted in fixed hierarchical roles (community partners/practitioners/academics) developed model approaches for better access to migrant communities jointly, and if possible, with shared decision-making power. From the perspective of the PaSuMi project coordinator at DAH, I (D.S.) reflect on the changes in relationships and power dynamics, as well as on my own role in the project. From different contexts, I was familiar with both, addiction, and grassroots work with and for communities. Questions and reflections accompanied me throughout the course of the project: “Right now, I am very unsure whether I am at all suited for this task. I need a lot of structure, clear lines, goals, concrete measures. But that doesn’t fit with the participatory approach. I don’t know if I can accompany the projects competently.” (Note from my research diary on 14 August 2017).

#### 3.2.2. Formation of Research Teams: Relationship Building and Change of Perspective

A first challenge was the formation of the local project teams because of the positionality of the participants, who were situated in different levels of hierarchically structured power dynamics. Some of the community partners involved were clients of the facilities while others were key persons of specific communities, independent of the facilities. Many of the participants had a history of migration some were drug users, sex workers or persons with experience of poverty. These characteristics often led to discrimination and exclusion. Central and continuous components of the team building process were therefore the development of trust and forming of relationships. Other important factors were sufficient time and resources, sensitivity, knowledge about and openness to different realities of life, and already established trusting relationships among the practitioners in the facilities. In some cases, contacts to community partners were established through other institutions or peer recruitment which, according to international literature, is a useful approach in participatory processes [27].

From my perspective, participation is fundamentally based on the assumption that nobody knows everything. The Canadian HIV/AIDS Legal Network for instance states, that persons who use drugs know best what works in their community. Furthermore, all persons should have the right to be involved in decisions affecting their lives [28]. Accordingly, I understand expert knowledge from lived experience as equal to the professional expertise of practitioners and the methodological expertise of academic researchers. Communities are thus equal contributors to the knowledge production process [29]. This challenged familiar roles in the PaSuMi project and required a change in the attitude towards collaboration with research partners. Through the interaction of different bodies of knowledge, the PaSuMi project tried to influence structures, to create new knowledge among all participants, to produce results relevant for communities that could be applied in programs and practice directly, and to build networks [30]. To what extent this has succeeded is probably assessed differently by each person involved. “How can we create free spaces within the given framework and to what extent is what we do really participation?” (Quote from a staff member of a facility, WSII on 17 June 2017) was a question that accompanied us throughout the process. I observed individual moments of change enabled by participation in the PaSuMi project, which are described below.

#### 3.2.3. Everything Is Different in Participatory Processes

Some of the eight local project coordinators not only faced the challenge of taking the PaSuMi project forward despite their own skepticism or insecurities concerning the participatory approach and in addition to their routine tasks in the facilities, but sometimes also had to work against resistance from within their organizations. For example, a coordinator had a tough time explaining travel and accommodation costs for participation of community partners at the regular joint workshops to the management. The management was not used to practitioners and clients in the role of community partners working as partners in a project. Another new aspect of the PaSuMi project compared to previous projects was that it was not only a matter of implementing specific prevention measures, but that the joint path that led to achieving this implementation, for example the formation of project teams and intervention planning, was an equally important part of the process. Some processes took longer in the PaSuMi project than in other projects and at first glance had little to do with the field of activity of the participating institutions or with direct addiction prevention. New methods were used and actions such as parties, a visit to an exhibition or a film shoot were implemented. Often, the initial focus was on confidence building, team building and skills acquisition, without addressing the issue of addiction prevention. In the first year of the project, participants increasingly allowed processes in PaSuMi to proceed differently than usual. It became apparent that participants discovered participation in the sense of a collaborative process as something that, due to changed relationships, opened new spaces for encounter and provided a previously unavailable insight into the lived experience of the community partners (and, perhaps that of the practitioners, too). “I need a lot of patience, but I’m really into the project”, remarked a local project coordinator (project visit on 10 October 2017) after his project team was established and ready to act.

Familiar ways of working and hierarchies changed. As a result, community partners began to use their decision-making power to help shaping the research process and determine what benefits they would derive for themselves from the project. This required a high degree of flexibility and openness on the part of practitioners and researchers regarding their usual ways of working and claims to power.

The moment when community partners approached me, took the research process into their hands, and assigned tasks to me, such as arranging a translator or organizing a meeting, was a key moment for me as project coordinator. I interpret this as a sign of successful participation. The relationships and trust that emerged within the project provided the appropriate space for this. To this day, some of the community and practice partners from the PaSuMi project are friends, contacts in my practical work and research partners in other research projects. Relationships established in the PaSuMi project, also among the community partners, continue to exist in various forms after the end of the project.

I was also confronted with my own claim to power, which I illustrate with the following example. One of the local projects included me in most of the steps of project planning and implementation at the beginning of the PaSuMi project. As the network on the part of the community grew larger and larger, certain activities, such as workshops or a meeting with members of the German parliament, took place without my knowledge. For me, this initially appeared to be at odds with the close relationship we had established, and certainly with my role as coordinator, who always wanted to be informed about everything. I had to learn that my discomfort had to do with control and power. The fact that the community partners did not let me participate in everything had nothing to do with me personally, but with the fact that my participation in some areas was not required. In retrospect, I feel that this was exactly my contribution to participation: to support the community partners where they needed it, and at the same time to accept that our collaboration was shaped together and that they decided what they shared with me and what not.

Similar experiences were also reported from the local projects. For example, in some project teams, the community partners requested various trainings because they wanted to learn how to give presentations and front events with confidence. Local PaSuMi project meetings often took place in the evenings and on weekends and thus outside the practitioners’ usual working hours. Without this flexibility and consideration of the community partners’ commitments in language courses, school and jobs, the meetings could not have taken place. Flexibility was also required in communication: in several local projects, communication took place mainly via WhatsApp at the request of the community partners, a channel not previously used in the counseling practice of the corresponding institutions.

#### 3.2.4. Complex Needs

Local PaSuMi project coordinators supported their community partners in writing job applications, finding functional clothes for internships and accompanied them on official visits. This was also part of the participatory research process, even if these activities had nothing to do with the content of the PaSuMi project and sometimes took place outside of regular working hours. Persons who do not have a secure residence status, whose rights are restricted by migration and asylum policies, and who are stigmatized and discriminated by society’s normative ideas and racism have complex needs that, even if they lie outside the research interest, must be taken into consideration in participatory research. The clear distinction between research, social work, and private engagement that exists in more conventional scientific approaches becomes blurred in participatory research. All this is an expression of a kind of relationship that does not exist in other forms of research.

Key challenges in the PaSuMi project were my own expectations of the participatory approach, the desire to involve the entire PaSuMi team in all steps equally, and the concern about making decisions without sufficient involvement of some stakeholders. This made some processes complicated and time-consuming. I had to understand that some persons exercised their right not to participate, had other priorities than answering an email, some might not even have an email address, I did not speak all the languages of the persons involved—in short, I learned that participation by everyone in decision-making could be aimed at, but it was not always possible. In conclusion, I think the PaSuMi project can contribute to the discussion of whether a lack of privilege is a barrier to participation in participatory research. I assumed that participation in the PaSuMi project was not impeded due to a lack of privileges, but due to specifically identifiable structural barriers, such as migration-related legislation, structures, and procedures in institutions and in research that had been established over many years, and due to established framework conditions in individual projects. However, through reflection and flexibility in their own approach, all institutions were able to find a way to win community partners for the project. What has become of the relationships that have grown over 2.5 years is known only to those involved individually. I know of three cases in two participating agencies where community partners have formally become colleagues. From some I know that they were looking for community support for the time after the PaSuMi project to continue PaSuMi contents independently. In addition, new networks were formed. As I write this, I wonder how all the persons involved are doing and how, in retrospect, they assessed the relationships, roles, power, and benefits (or harms?) of the PaSuMi project.

### 3.3. Shifting and Interweaving Perspectives in the Project “Back into Life-with a Power Wheelchair”: From Occupational Therapist to Research Initiator and Companion, by Author Tabea Böttger

#### 3.3.1. Introduction

The project presented here was conducted between 2017 and 2019 in the context of the author’s working environment, a long-term rehabilitation center for adults with severe acquired brain injury in Berlin. At the same time, the author took part in a part-time occupational therapy master’s degree program at the University of Applied Sciences and Arts (HAWK) in Hildesheim, Germany. The author’s aim was to address a practice-oriented issue in her final thesis and to involve former rehabilitation participations as co-researchers with equal status in order to implement the participatory research approach she had become familiar with in practice [31].

Many stroke survivors experience permanent limitations in their activities of daily living, including mobility. Some lose their usual walking mobility or are only able to walk a few steps inside their home without assistance. To be able to carry out activities outside the home (e.g., shopping, visits to the doctor, visiting friends) as autonomously as possible and without assistance, some rehabilitation patients are advised and provided with a power wheelchair. While intensive training with this assistive device takes place in different contexts (such as in public spaces) during rehabilitation, it is unclear whether individuals use it after discharge for mobility outside their home and to what extent it supports them in their participation in social life. Based on these considerations, the following research question, which I formulated on my own at the beginning of the project, was at the center of the project: How do persons with a severe acquired brain injury experience their community mobility in a powered wheelchair in the metropolitan area of Berlin and what changes do they want to initiate?

The entire research team (author, a colleague from the occupational therapy department from the rehabilitation center and five former rehabilitation participants) met in five group meetings over a period of one year (May 2018 to May 2019). Photovoice was chosen as a method to answer the research question [32]: five persons after stroke took photos of their living environment alone or with the help of others, presented them to each other in the meetings, discussed and analyzed them together with the aim of publishing their results to draw attention to their concerns. The photos and stories were first exhibited in the long-term rehabilitation center at the request of the co-researchers (January 2019), then in other contexts (e.g., scientific conferences, Occupational Therapy School).

#### 3.3.2. Relationship Building between Research Initiator and Co-Researchers

From the author’s point of view, the issue of relationships played a central role in the research project. The following hypothesis serves as a starting point to the discussion: Without the pre-existing, trusting relationships between the author, her colleague and the five co-researchers, this project would retrospectively probably not have taken place in this form.

All co-researchers had cognitive, communication and sensory motor impairments due to severe stroke (up to 12 years ago). So far, this group of persons has been hardly or not at all considered in traditional medical und participatory research [33,34]. Especially the severity of their cognitive as well as communicative impairments are mostly only considered as exclusion criteria, as these persons are, among other issues, considered incapable of giving consent and/or uncertainty prevails as to how their views can be captured despite their manifold impairments. This required ethical considerations and decisions on the part of the author before the beginning as well as during the entire research process. A positive vote was obtained from the Commission for Research Ethics of the University of Applied Sciences and Arts (HAWK Hildesheim, 3 April 2018). By communication impairments, the author means acquired speech and language disorders like aphasia and dysarthria [35]. Cognitive impairments are understood here primarily as impairments in the areas of attention, memory and executive functions, including deficits in self-awareness [36].

But what conditions are needed to actively involve these persons in a participatory research team on an equal basis?

From the author’s point of view, this first required researchers who, in addition to openness and flexibility for the research process, above all had specialist knowledge of the individual impairments and their relevance to everyday life. It also required experience in being able to adjust appropriately to these persons in communication in order to open up protected spaces for them in which they could express themselves verbally and nonverbally via gestures, visual material, electronic communication aids or written language despite their impairments. Internationally, initial participatory research with persons after stroke exists, but mostly involved persons with little or no cognitive and/or communication impairments [37,38,39].

#### 3.3.3. Common Background of Experience

The author and initiator of this research project had, through her professional background as an occupational therapist, both contact with persons with complex support needs and, through years of working in partnership in therapy, had become acquainted with and tested ways and means of enabling communication with each other. Thus, she took over the function of gatekeeper and, with her previous professional experience and contacts, enabled access to a group of persons often described as “difficult to reach” or “marginalized persons and groups” [40] (p. 52 translation by the authors). The author selected specific persons and contacted them in different ways, with whom she had worked to varying degrees of intensity during inpatient long-term rehabilitation in the last eight years. First, two former rehabilitants were contacted with an information letter in easy language, with whom the author still had sporadic personal contact. In a second step, these persons were able to name other former rehabilitants who were asked to participate in the project. In this way, the relationship is consciously considered in the recruitment of participatory co-researchers and thus in the composition of the research team. At the last group meeting, which was dedicated to the evaluation, two co-researchers confirmed that this aspect was quite decisive for their “participation”: “I participated […] because I also wanted to meet persons again” (Steven, 18 May 2019) “[…] and I was interested in (…) research (…). Daniel, Petra, (…) and Ines (counts with fingers and points to persons present). And then I say yes. Because I know them” (Berta, 18 May 2019).

At the time of invitation or cooperation in the research project, a therapeutic relationship no longer existed. But they received an invitation from a person known to them to participate as active co-researchers. The pre-existing therapeutic relationship established a basis of trust for further joint cooperation, as it was oriented towards the principles of client-centered practice [41]. Client-centered practice is characterized by a relationship based on partnership, which requires an active and equal involvement of the clients in decisions about goals and interventions of therapy, recognizing the individual wishes and needs as well as autonomy of each person (ibid.). Despite this demand for therapeutic collaboration, in the author’s view, a relationship of dependency remains, defined among other aspects by the traditional roles of therapist and patient/client. The desired cooperation in the research project should depart from these familiar role patterns and allow for a more liberal cooperation, for example, without entering into a therapy contract.

Another central issue was the chosen research topic “community mobility with a power wheelchair” that connected author and co-researchers. Within occupational therapy, the possibilities of this assistive device for independent mobility outside the home were discussed and, above all, intensive training sessions were conducted in public spaces in the various settings. Thus, the author, her colleague as well as the co-researchers had numerous shared experiences regarding the everyday challenges in using the assistive device and were thus able to draw on a common content basis/experiential knowledge.

#### 3.3.4. Experience in the New Role

Due to the pre-existing personal relationship mentioned above, there was a basic openness for the project idea among the people concerned, without knowing precisely what a participatory research project was and what their cooperation could look like. Taking into account the risk of exercising power and influence involved, the persons approached were given extensive and appropriate information (at several points in time and through various media and communication channels) about the nature of the research style and their associated active role. This initially included a personal conversation in their familiar home environment using written educational material in easy language and was carried out continuously at the beginning of the meetings in the sense of an ongoing assent [42].

The description of collaboration, joint research, and the attempt to initiate change spontaneously led to uncertainty and rejection among all persons asked. Fears were expressed of not being able to live up to the attributed competence of being active as equal co-researchers, as well as a feeling of powerlessness, of not being able to influence the prevailing social conditions. Further, very practical concerns mentioned for the planned group meetings included not having enough time due to busy daily schedules or not being able to reach the meeting place alone without assistance and much effort. The author finally succeeded in convincing the persons for a first joint meeting. The following factors were probably essential for this, which the author took into account in the organization: a centrally located and barrier-free room (and toilets) in Berlin, that was easily accessible by public transport, assistance in organizing the time window and the route, the use of various communication channels and, in part, appointment reminders. In addition, the author was clear in the discussions that this project was a joint effort that could also fail. Even before the first meeting, it became clear that the “researcher’s” area of responsibility encompassed much more than the preparation and design of the content of the meetings and what significance the aspect of accessibility also had in the specific research context.

The author constantly expressed her ideas of equal cooperation and at the same time had to get involved in a process that could only be controlled to a limited extent. The participating co-researchers repeatedly decided for themselves to what extent they would contribute to the project-or not.

This project focused on mutual encounters and learning as Altrichter and Gstettner [5] (cited in Heinze 1986, p. 8) stated. Co-researchers were integrated into the research process on an equal basis to jointly explore their experiences and to initiate change. This meant that the author had to leave familiar paths and slip into the role of a researcher, which was new to her, and fill it with life, especially in contact with persons she had already supported as a therapist in another setting. In the process, it became apparent to her that it could profitably apply many competencies and ways of acting from her professional practice. At the same time, it became clear again and again in the research process that the initially ambitious goals of “equal inclusion” in the “joint” project needed gradation. Thus, in the first two meetings, the co-researchers understandably did not speak of their or our project; after all, they had not initiated it, but had been invited to “participate”. Accepting this was difficult at first. Patience and an open discussion during the meetings about the own claim of the “joint project” beyond the writing of a qualification paper, however, made clear to the persons concerned their status in the project. At the same time, their tasks and responsibilities increased. For example, in the later course of the project, a lengthy discussion arose in the research team about where the first exhibition and thus presentation of the results should take place. The co-researchers stated their different opinions, and the author and her colleague supported the decision-making process, especially through facilitation.

Nevertheless, it was important to accept that each person could and wanted to participate with different intensity and that therefore there were differences in participation that should not be negated or concealed. Disclosing the unequal preconditions, such as different cognitive and communication abilities, and demands prevented egalitarianism. In this participatory research project, it was important to allow for different scopes of co-design, situated in an area of tension between enabling influence and avoiding excessive demands.

Many things were new for the author: she learned to reflect more consciously on the characteristics of her role as a therapist and to take a step back when, for example, behavior was shown that was classified normatively as negative (such as excessive consumption of stimulants such as sweets, nicotine and alcohol), and to address this only when, in her view, it influenced the group work process negatively or when other co-researchers, including herself, felt disturbed by the behavior. However, this confrontation was not only about her professional role as a therapist but involved a conscious reflection on her own personal attitudes towards socially prevailing norms and values. It became clear once again that persons with disabilities are still primarily seen as weak and in need of help and less as experts in their own lives. This was also reflected very clearly in the goals the participating co-researchers stated for the project: on the one hand, they wanted to inform their personal environment, friends, and family, about their own lives–especially the challenges and discrimination they experience every day. Furthermore, they wanted to show a different image of persons with disabilities in general and contribute to the reduction of prejudices, fear of contact and discrimination.

Nevertheless, the behavior and action on the author’s part were shaped by my previous experience in the therapeutic context, which was ultimately helpful in many situations, such as the spontaneous assistance with a toilet transfer, which was proactively requested by the person concerned. To develop a better understanding of the living environment of the co-researcher, it was necessary to allow unexpected situations to occur spontaneously that did not primarily contribute to the goal of the project but were in the interest of the co-researcher. This approach was perceived as enriching for the joint process. For example, the author twice went out to dinner with individual co-researchers in a restaurant at their request after a joint excursion of photographing. Thereby she gained very vivid impressions of their perspectives and thus gained more knowledge as a researcher and practicing occupational therapist as well as an individual person outside these contexts and used this knowledge for critical reflection. During the entire research process, it became clear that there was space for both roles of the author, as they were relevant depending on the situation and at the same time, there was an increase in knowledge for both activity profiles.

#### 3.3.5. Enabling New Spaces

Even though the project was initiated and led by the author, no new dependencies were created [5] (p. 69). Instead, a temporary space for exchange, getting to know each other and mutual appreciation was created, which had not existed before. The co-researchers lived in various districts of Berlin, in their own apartments or shared apartments, usually with little or no social contact outside their regular help or family system. During the group meetings, they were able to experience that they were not alone with their everyday problems. They supported each other at the meetings when, for example, a person with aphasia and/or apraxia of speech could not find the words. At the beginning of the meetings, they greeted each other warmly and inquired about each other’s well-being. Some exchanged contact addresses. The sense of belonging to the project and the feeling of “we” among each other increased significantly during the meetings. To what extent the exchange among each other was continued after the end of the research project is not clear. Some co-researchers meet irregularly in other contexts (regulars’ tables, parties at the former rehabilitation facility). The author has kept loose contact to the co-researchers, in case of requests for project presentation she forwards them to all co-researchers.

Each co-researcher decided for him/herself how long he/she would participate in the research project: for example, one co-researcher decided after the first exhibition that this was the end of the project for her and that she did not want to participate in further events. The author received further offers to make the results available to a wider audience in the form of exhibitions. She decided to do this only with the consent of the co-researchers and their participation (at least one person), so that it would remain a joint project also in the presentation and not become a pure end in itself.

The personal experiences presented here intend to illustrate that a participatory research project with persons with diverse impairments (cognitive, communication and mobility) rests on many preconditions and that pre-existing personal relationships and specific professional competencies play an important role in the implementation of such a project. At the same time, persons interested in participatory research are invited to engage in the critical reflection of their own competences in the preparation and during the research process. Furthermore, prospective scientists from the therapeutic and social professions are encouraged to consciously use their professional background for this research approach.

### 3.4. Towards Participation for Nurses in the Realization of New Professional Roles in Community Care: Workshops with Stakeholders as a Pilot Project by the author Doris Arnold

#### 3.4.1. Introduction

The realization of participatory research projects in which participants take an active part in the research process as co-researchers depends on many conditions. This sums up my experiences in a research project on the development of expanded professional roles for nurses in community care for persons with dementia, which was carried out in the final phase of the project “E to the power of B—Nursing and Health”, funded by the German Ministry of Education and Research. The original plan was to conduct a participatory project with participation of research participants as co-researchers [43]. However, we found that participation in research requires time and resources that were not available at that point. Thus, we carried out a pilot project to create the conditions necessary for the realization of meaningful participation in a subsequent participatory research project.

I acted as primary researcher in the pilot project, who planned the research process, conducted workshops employing focus groups together with colleagues from the project “E to the power of B—Nursing and Health” and interpreted the findings. The workshops aimed, firstly, at the design of a new professional role for experienced nurses who are qualified through a certificate program on extended community nursing practice for persons with dementia, and secondly, the identification of ways for financing these extended nursing tasks. Participants of the workshops were experienced nurses, who had attended the certificate program and other stakeholders in community care and health insurance.

In the following, the context of the project “E to the power of B—Nursing and Health” is initially described. The limiting circumstances that led to the decision to abandon the plan to use a participative approach and to implement the pilot project instead, consisting of three focus groups and its differences to participatory research are explained. Finally, the intended subsequent participatory project is briefly introduced.

#### 3.4.2. The Background: The Project “E to the Power of B—Nursing and Health”

The certificate program “Care Strategies and Psychosocial Support for Living with Dementia at Home” [44] is as a scientific continuing education program developed and tested in the project “E to the power of B—Nursing and Health”, which was funded as part of a government initiative aimed at the development of continuing academic education in German Universities. Academic education in German nursing still is very poor in all sectors, with only about one percent of nurses working in direct patient care in German teaching hospitals holding academic degrees [45].

The focus of the project was to develop, test and sustainably establish scientific continuous education for experienced nurses, who represent a non-traditional target group for academic education in Germany. An important argument for the approval of the funding for the originally planned participatory project was that the implementation of a new professional role for graduates in community nursing may contribute substantially to the sustainability of the certificate program. The development of the certificate program was based on the results of a comprehensive assessment of needs in community care and in further education for nurses [46,47]. It qualifies professionally experienced nurses to systematically evaluate, understand and answer the needs of persons with dementia who show so-called challenging behavior, as well as the needs of their family caregivers, based on a person-centered approach [44]. Thus, on one hand, the results of the needs assessment point to the need for extended tasks in caring for persons with dementia living at home, that should be carried out by appropriately trained nurses. On the other hand, these tasks had not yet been realized in the practice of community care and had not been financed accordingly. Given this background, the aim of the research was to identify and overcome challenges for implementing expanded nursing care for persons with dementia by drawing on the knowledge of nurses experienced in community nursing, who attended the certificate program, and that of other relevant stakeholders in community nursing and financing of health care. However, even though initially a participatory research project aiming at the implementation of expanded community care for persons with dementia appeared to be the most suitable approach, it quickly became clear that the tight time span of the overall project as a third-party funded project severely limited the possibilities for more extensive participation in a joint research process. For example, due to the limited time for appropriate advertising, relatively few nurses working in community nursing services could be recruited to participate in the piloting of the certificate program on dementia in community care. Therefore, it was not possible to involve the community nursing services in which the participants were employed as care professionals in the project to a sufficient extend. The original plan intended the establishment of local working groups, in which participation of nurses who had attended the certificate program and other local stakeholders as co-researchers in focus groups should be facilitated [43].

In the ultimately realized pilot project, only an involvement of participants was achieved, and its goals were limited to gaining more knowledge about what new, expanded professional roles and tasks for scientifically trained nursing professionals in community nursing services could look like in the context of dementia care and how this can be financed as a first step. This should be followed in a second step by a participatory project aimed at the implementation and evaluation of this expanded nursing role in community care, which is planned as a separate project.

#### 3.4.3. The Pilot Project: Opportunities for Participation and Barriers to Participation in the Research Process

The pilot project finally comprised three workshops with focus group discussions with different groups of relevant stakeholders: (1) graduates of the piloting of the certificate program, (2) nursing service managers (PDLs) and managing directors of community nursing services, and (3) experts on the financing of community nursing care [48].

The project used qualitative research methods that allowed participants to be involved in some research activities but remained in the realm of involvement and did not allow for active co-operation in research activities and decision-making as co-researchers. Thus, the method of data collection in focus groups [49,50] offered the participants the opportunity to set their own priorities. As results of the individual focus groups, summary protocols were prepared, oriented on the procedure of qualitative content analysis [51]. Protocols were sent to participants with an invitation for feedback, so that they had the opportunity both to check on their contributions represented in the data and to make additions. In the second and third workshops, the available findings from each of the previous workshops were presented at the beginning. In this way, the cyclical process of participatory research was to some extend included in the procedure [40,52,53].

As mentioned, active participation of participants as co-researchers or shared decision-making in the research process could not be implemented in the workshops [14]. Participants did not participate actively as co-researchers in data collection and analysis and were not involved to any significant extent in the publication of the results. Most participants of all three workshops contributed with their respective valuable expert knowledge to the research process, as is usual in the context of qualitative research, and only a few persons participated in feedback. This gave them the opportunity to read and check their own contributions to the discussion in the form of “member checks”.

However, some participants did mention a direct interest in the planned implementation project: for example, one nurse who had attended the certificate program was interested in implementing such a project in her community nursing service, and individual nursing managers expressed a special interest in the project. In this way, important contacts in the field were made, which can be followed up in the implementation project. In the workshop on financing, further specific considerations on the subsequent research project were discussed with the representatives of a nursing care insurance company and of a provider of community nursing services. At their suggestion, the planned model project will include two subprojects: The participatory implementation project and evaluation project that should provide evidence for the efficiency of the new nursing role. Thus, this also created substantial contributions that provide a sound basis for the future second step.

#### 3.4.4. Evaluation, Outlook, and Thoughts on Participatory Research

In retrospect, the implementation of the workshops with focus groups within the limited conditions of a third-party funded project seems justified. On the one hand, it was necessary to forego the creation of additional possibilities for the participation of participatory researchers. On the other hand, meaningful first drafts for an extended professional role of a “dementia consultant in community care” and its activities were developed with comparatively little effort. Most significantly, a first proposal for financing these extended nursing tasks was drafted together with the representatives of a nursing care insurance company. It was particularly important that the cooperation with the respective participants at the workshops was as transparent and respectful as possible. We asked the respective actors to provide their expert knowledge for a specific goal and offered them opportunities for feedback and thus also for control over their data.

I have continued to maintain contact with three participants, who have expressed a special interest in further cooperation: one nurse who attended the certificate program, one contact person in a nursing care insurance company and one representative of a provider of community nursing services. To these persons, I have sent publications that I wrote about the project, and hope that the plans for the envisaged future participatory implementation project can be realized, in which I would then like to work more intensively with them.

The results of the workshops as a pilot project represent nothing more, but also nothing less than important prerequisites for the planned participatory implementation project –provided suitable funding can be obtained. Then, the research team would work in close cooperation with individual community nursing services. Nurses working in teams of community nursing services would be trained in the next certificate program offered and subsequently work together with the researchers as central actors and co-researchers. In doing so, they would lead the research and development process in implementing this new nursing role in their community nursing services in collaboration with other key stakeholders. Most importantly, among these stakeholders should be persons with dementia and their caregivers acting as co-researchers, as their health and wellbeing is in the focus of the expanded nursing tasks of the future consultants for dementia in community care. In this future project context, active participation of all co-researchers in all parts of the research process on an equal basis is not only plausible but is also indispensable.

However, there are still some hurdles to overcome before that project can be realized. The biggest of these is the acquisition of funding, for which it is helpful to be able to refer to the results of the pilot study. On the other hand, community nursing services whose managers support this project must be recruited. Most importantly, individual nurses who work in these facilities, and who are interested in participating in the certificate program and in becoming co-researchers in the participatory research project must be found.

In the planned participatory project, empowerment as an emancipatory moment of participatory research is not intended for clients of health care in the first place, even though their wellbeing is the aim of the nursing care in question. Rather, that project is an example of participatory practitioner research, and nurses as professionals and employees of community nursing services are central actors and therefore, they are the persons who are to be empowered to change their professional role as nurses. As mentioned above, persons with dementia and their family caregivers, should also be encouraged to participate actively as co-researchers. This means that their voices should also be heard and their perspectives on problems should be made visible, but they are not the in the focus of the project. Rather, the implementation of a new professional role for scientifically trained nurses is closely linked to an organizational development process [3] within community nursing services as service providers.

### 3.5. Approaches to Participation in the DIPEx Project on Illness Experiences Based on Narratives from the Patient’s Perspective by Author Andrea Glässel

#### 3.5.1. Introduction

On the one hand, the implementation and success of participatory approaches in research projects holds great potential in terms of joint development and learning from and with each other and shaping the research process. On the other hand, these participatory elements that hold promising potential pose difficulties to implement in research practice and can sometimes be demanding for researchers and participants involved. The participants in the project are in this case, patients. For them, as well as for the researchers, the research process, and the building of the relationship between researchers and participants, are characterized by different challenges and expectations. Based on a current research project, which focuses on experiences of and perspectives on the experience of health and illness by means of narrative interviews, this example would like to show experiences from the perspective of the researchers on participatory elements within the shared process experience and relationship design. In this context, individual elements of participatory research are integrated at various points within the research process. This contrasts with an approach to participatory health research, as described by the ICPHR with the goal “to maximize the participation of those whose life or work is the subject of the research in all stages of the research process” [15] (p. 6).

#### 3.5.2. Presentation and Introduction to the “DIPEx” Project

The Swiss project DIPEx.ch is short for Database of Individual Patients’ Experiences and is a national database for individual patient experiences based on narratives from the patient’s or affected person’s perspectives on health and illness (www.DIPEx.ch, (accessed on 26 July 2022)). This approach of building a national database was developed at Oxford University in 2000 as a pioneering project with direct patient participation, in which patients and their experiences were at the center as experts and thus given a public voice for the first time. Since then, international, and German-language participatory research and the possibilities of empowerment, participatory co-design, on the one hand, and the reception habits of users, on the other hand, have developed further due to digitalization (Web 2.0/social media), as shown by approaches to experienced-based co-design procedures and integrating a participatory approach [54].

The database contains a systematic and methodological collection of narratives on the subjective experience of illness and health from the perspective of those affected, as patients and/or as relatives. The guiding questions of the project are: “What do persons experience when they suffer from a serious illness or health condition? What experiences do they have as patients in doctors’ and health professionals’ offices, or hospitals? How do they cope with their everyday life with the restrictions caused by their illness, new functioning and changed health situation? What kind of support do they find helpful and where would they like to see more services? What problems, questions or answers do patients have that doctors, therapists, nurses or researchers may not yet have in mind?” and other questions.

The DIPEx project was designed to present excerpts of real patient narratives on the topic of multiple sclerosis (MS) as an example, on the website in the form of video, audio, or text excerpts. This includes a systematic collection and analysis of interviews about the individual experiences, descriptions of the emotional experience, allows insights into the biography, as well as into the everyday experience, or to different forms of treatment and more. In addition, information on self-help groups and other materials, including links to answers to participants’ questions, are provided.

At this point, according to Bergold and Thomas [55], the association to participatory research becomes apparent. Without participation in research, “DIPEx” cannot be implemented. It takes place “together with the persons directly affected and aims at reconstructing their knowledge and skills in a process of self-understanding and empowerment. The majority of these are vulnerable groups of the population whose perspectives and voices are otherwise rarely included and who themselves hardly have the opportunity to justifiably introduce and assert their interests” [49] (pp. 7–8, translation by the authors). To be vulnerable because of the disease process, which demands its very own attention, which requires different or new daily structures, under which one’s own participation and involvement raises the question: As a patient and/or family member, do I want to belong to this group at all and/or actively confront this difficult topic?

DIPEx Switzerland (www.dipex.ch, (accessed on 26 July 2022)) is an interdisciplinary group of researchers at the Institute of Biomedical Ethics and Medical History (IBME) of the University of Zurich and its partner the Zurich University of Applied Sciences (ZHAW) at the Department of Health Sciences. It is both integrated into a Swiss-wide network of partners in health care and a member of the international umbrella organization DIPEx International (www.dipexinternational.org, (accessed on 26 July 2022)). The network currently includes 14 countries with national websites for the UK (www.healthtalk.org, (accessed on 26 July 2022)), Germany (www.krankheitserfahrungen.de, (accessed on 26 July 2022)), or www.healthexperiencesusa.org, (accessed on 26 July 2022) in the US and others. These national databases follow the long-established qualitative research methodology set out in the Handbook of the Health Experience Research Group (HERG), Department of Primary Health Care, University of Oxford. The publication of real patient narratives in DIPEx follows the HON criteria (Health on the Net Code) and as a freely accessible resource, it forms an important support for patients, relatives, clinicians in training as well as teachers in the health professions and for research. Studies could show that patient narratives can have an impact on the individual experience of illness, but also on health policy, and on affected individuals reconsidering their own approach to illness, as when reading the German DIPEx website [25,56,57,58]. Drewniak, Glässel, Hodel, and Biller-Andorno [59] were able to show that access to individual experience is perceived as relevant support and can positively influence the empowerment of patients regarding disease management. The findings of Fadlallah et al. [60] suggest that, in addition to individual inspirational and empowerment effects, illness narratives can also serve as educational and awareness-raising tools of the political discourse on illnesses or health. The strength of individual patient narratives is also invoked by the 2018 European Report for Health, entitled “More than Numbers” (WHO). It calls for individual patient narratives to serve as illustrations alongside statistical information and facts, to strengthen the understanding of health, and to be specifically included in the formation of health literacy in the population [57].

#### 3.5.3. Experiences on the Implementation of Participatory Elements within the DIPEx Project

The DIPEx project follows a qualitative study design according to the HERG manual, which DIPEx members are taught during mandatory training. This training is a prerequisite for the research groups to obtain a national DIPEx license. The DIPEx Switzerland project is conducted according to the principles of the Declaration of Helsinki [61] and received Swiss-wide approval from the cantonal ethics committee with the number BASEC Req-Nr. 2018-00050.

The procedure for the survey of lived experiences of patients with MS was methodologically based on a phenomenological approach using narrative and semi-structured interviews, which are based on the sampling strategy of maximum variation. Possibilities for limited participation or involvement of the patients in the research process become apparent in this approach, such as the fact that the participants are given a lot of space to set their own priorities within the narrative interviews. Some of the interviewees saw this as an opportunity to stimulate conversation and to be able to recount their experiences in depth from their own perspective. At the same time, it was precisely this open form of narration and of setting one’s own focus that was experienced as challenging by other interviewees. These persons found the guiding questions by the researchers to be relieving and helpful. They helped to structure the narrative of their experience and their thoughts along the questions so that they could communicate more easily. In this interactive event of the first personal contact and encounter, the researcher is challenged to establish a relationship level conducive to conversation as a basis for the interview immediately and to convey an open and approachable attitude.

To create an atmosphere that was as natural as possible and that stimulated the conversation, the interviews were conducted in everyday language and recorded with video and audio. Verbatim transcripts were prepared using established rules for transcription as a basis for evaluating the interviews. Transcripts were sent to the participants by mail with the request for feedback, so that they had the opportunity for transparency about their data and statements as well as for additions. From the researcher’s perspective, this type of participation or involvement in the research process revealed a great deal of ambivalence among respondents regarding (a) reading their own story and experience of illness and (b) reading their own linguistic expression. Reliving one’s own story again and yet anew via the written discussion requires a deeper engagement with oneself. The personal state of mind and feeling in this situation of narration is due to the moment and to the atmosphere of the conversation with the interviewer. It thus also acquires a temporal and very personal, emotionally individual, even partly intimate dimension. Reading one’s own transcript is not only about reading and correcting a statement or coherence of content. It is also a moment of reflection and awareness of emotional vulnerability and deep illness reflection [56].

Reading one’s own language, formulation, and expression in everyday language or in a modified form of one’s own dialect partly triggered a great discomfort, a defense, a confrontation with the statements produced in spontaneous speech, the text. Despite assurances by the interviewer that it was normal for spontaneous speech to be grammatically incorrect, that a spoken word did not correspond to a written word, this discomfort arose in some of the participants. The same applies to the unfamiliar text format of a transcript with special characters, although the researchers announced in advance the rather special text format, which does not conform to everyday speech, compared to linguistically polished texts in journals or books. As shown by Lucius-Hoene et al. [56]: the transcript aroused astonishment to the point of horror in the patients about the unfamiliar confrontation with what was said in the text’s own language. The statement of one patient: “When I received the first written excerpt from what I had said, I was first of all shocked at myself and said, ‘Is that how I present myself? What did I say? The way it was written, that’s how I speak? That is terrible and that is not possible” [56] (p.199).

The feedback of the data to the interviewees is intended as a participatory element in this process and is perceived as a balancing act from the researcher’s point of view. It requires an examination of one’s own research contribution and a balanced relationship of closeness and distance between the researcher and the participants. Participation addresses the relationship between the researchers and the participants, including the researched objects about which they generate knowledge, and that new knowledge already emerges in relation to the discussion of this relationship within the research process. The design of this relationship level influences the handling of the data and information and thus in turn the participation and further development in the project.

As already described in Altrichter and Gstettner [5] (p. 68), the claim of participation with a symmetrical dialogue situation and cooperative research of researchers and participants in practice is presuppositional and is only conditionally achievable due to limited framework conditions in research to invest in this relational work. This form of relational work also often does not meet the expectations of research from the perspective of both researchers and participants. In our experience, participants in DIPEx do not expect researchers continuing in the role to maintain contact after the interview until the transcripts are received or the experiential posts and narratives are unlocked on the website. In the run-up to the interview, detailed information about the project was provided and the subsequent steps described were outlined. Nevertheless, according to previous experience, it was only partially possible to sufficiently convey the feedback of the data to the interviewees or to alleviate or even dispel concerns about it. This means for the research practice that cooperation and a form of co-production is in principle intended. In the design of the roles, however, this co-production in the sense of a participatory procedure usually succeeds only inadequately with rather one-sided role differentiation. The imbalance consists in the fact that the researchers have more experience about research-related circumstances and do not question the unusual format of a transcript further but regard it as part of the methodological procedure. At the same time, researchers themselves have usually never or rarely been in the situation where their own statements were written down in everyday language and had to be proofread by them with the knowledge that parts of them would be published. This creates an imbalance that does not promote true participation in the sense of sharing. In principle, mutual learning is possible, but the gain in knowledge is not visible to the participants in the same form or elaborated accordingly within this rule-governed research process [5].

#### 3.5.4. Thoughts on Participatory Research and Outlook

With a self-critical view of the participatory element described in this project, the “intended level of participation” of the participants was rather limited. Sharing the experience: Yes, and gladly! But more participation is not needed. In this case, the intended project design with the goal of a high level of participation demanded more from the persons involved than they were willing or able to contribute, and thus also made limits clear.

In the further research process of the DIPEx project, other participatory elements can be identified, such as the advisory group, which is composed of different persons including patients. Based on this abbreviated insight into the DIPEx project, this approach cannot be further demonstrated. With reference to the chance of improving participation within the DIPEx project, the possibilities for an improved participatory and co-creative co-design of the personal contributions on the website are currently being investigated within the framework of a participatory project, to prospectively strengthen the project through participation in the form of participation and to work out its potential in a co-creative process.

## 4. Discussion

Based on the four examples presented, our central concern is to identify, name and reflect on the numerous factors that influenced the course of the concrete projects in their respective contexts in order to further the development of participatory research. In the following, we discuss these factors in relation to the shaping of relationships and the level of participation achieved and highlight both the various success factors and the challenges in participatory research processes.

### 4.1. Building Trusting Relationships in Participatory Research Processes

The experiences and reflections from the four projects described here show that the establishment of trusting relationships between academics as researchers and other research participants is a central element and a foundation for the success of a participatory project, as is shown in other research about CBPR [13]. At the same time, it is evident that this process has taken up different amounts of space, time, and weight in the four projects. While in the projects PaSuMi (Navina Sarma) and “Back into Life” (Tabea Böttger), trusting relationships between and among the research participants and the research initiator or the project coordinators existed in part already before the start of the project, these were initiated in the “Workshops” (Doris Arnold) and DIPEx (Andrea Glässel) only during the project. In the descriptions of the PaSuMi and “Back to Life” projects it becomes clear how access to the desired target group, the persons, and institutions relevant to the research in each case was facilitated by already established trusting relationships. Thus, various migrant individuals and groups and persons with severe stroke could be successfully involved, while in the “Workshops” the lack of existing relationships and networking also limited access to nurses in community nursing services and thus their recruitment for participation in the research project. Building these relationships would have required significantly more time and effort than was feasible under the conditions of the externally funded project. This observation applies to DIPEx, where in-depth work on building relationship with participants was only possible to a limited extent given the external condition of a one-off interview. This underlines limitations posed by deadlines of funded projects that cause too little time and resources to facilitate more than tokenistic participation in applied health research mentioned by Green and Johns [9].

In the context of CBPR, the need to work and reflect on the positionality of researchers when building trust with members of marginalized communities has been shown [12,19]. Similarly, despite partly pre-existing relationships in the PaSuMi and “Back into Life” projects, the new kind of cooperation as partners had to be negotiated to overcome problems with positionality in previously existing hierarchically structured power relationships in the respective work contexts. This grew gradually during the project, primarily through sufficient sensitivity, knowledge, and openness to the different realities of life and experiences made, as well as in encounters outside the joint working meetings. Both authors describe how opportunities were created for activities desired by the research participants outside the originally planned joint working meetings repeatedly. Thus, also concerns of the persons involved, which at first glance had nothing to do with the goal of the research project, were considered. According to Navina Sarma and Tabea Böttger, formulating and dealing with (personal) needs of research partners was part of the participatory process and supported trust building.

To what extent are these activities to be understood as a necessary part of participatory research? Or, to what extent do they reveal to us, the “primary researchers”, in which everyday issues individuals have perhaps received too little support so far and/or that they do not know where they should turn, for example, to look for work clothes for internships? Do existing grievances show up more clearly, or at all, in participatory research projects due to the existing trusting relationship? And how do the project initiators deal with this?

Our experiences have led us to enter into participatory projects with the knowledge that, precisely because relationships and trust are there, situations will arise in which we will act as individual persons and not as scientists. To which extent and how much private time is invested for this purpose is a matter for discussion and individual decision.

The PaSuMi and “Back to Life” projects make it clear that, in addition to pre-existing relationships, the number of joint working meetings over a longer period contributed significantly to relationship building. Neither was the case in the other two projects: in the “workshops” there were one-time meetings with the respective group to conduct the focus groups; in DIPEx there was also only one meeting each to establish contact and conduct interviews—partly initiated while planning and arranging the interviews—by e-mail or telephone or through mediating contact persons. Based on the projects described above and their progress, it becomes clear that a person initiating the research with already existing long-standing, close contacts to a group of co-researchers has much more far-reaching possibilities to establish sustainable relationships in the research process. The form in which relationships are established also has to do, among other things, with the question of the form in which the methods used are (or can be) designed to be participatory.

Another influencing factor is our own understanding of our roles as researchers or project initiators, which we bring with us and live out in the projects. Some of us see ourselves as impulse givers, activists, or perhaps see themselves as practice partners, or move back and forth in double roles between therapy and science. The experiences in the PaSuMi and “Back to Life” projects show that the collaboration with the respective research participants can result in a sustainable relationship if we as academics disclose the different preconditions in the research team and at the same time focus on the necessity of the different backgrounds of experience for the common interest of creating knowledge. The disclosure of different preconditions and interests can result in the consequence of distributing tasks and responsibilities differently in the project team depending on the research step and allowing non-participation. For example, Tabea Böttger in “Back into Life” wrote down the results of the group discussion alone in a text form based on the jointly determined categories from the evaluation meetings and submitted these to the persons involved for proofreading before publication in her qualification thesis. Just like Doris Arnold in the “Workshops” and Andrea Glässel in DIPEx, she did not receive feedback from all persons involved in the desired Member Check. This also became apparent in the DIPEx project based on the question: May, or should or even must the standardized procedure of the Member Check by counter-reading the interview transcripts, as required in the method manual, be deviated from, if it does not correspond to the individual needs of the individual persons, as suggested in the contribution by Andrea Glässel? The method manual is seen as a helpful, quality-assuring basis for a common approach, especially for a cross-national research project. At the same time, it should open space for project-related experiences, so that on both sides, the researchers as well as the participants, positive or at least satisfactory experiences are not subsequently overlaid by negative experiences, which the researchers can only influence to a limited extent, as they cannot satisfactorily accompany the process of reading and dealing with the transcript in certain situations. It becomes clear that a standardized approach without allowing for individual nuances cannot meet or even contradicts the demands of participatory research.

### 4.2. Participation in the Area of Tension between Time and Dosage

At the end of the project, the question arose for all four projects about to how to deal with relationships that had arisen in the context of the research project.

With whom do I maintain contact and with what intensity? What criteria do I use to decide this matter? Does it play a central role whether, at the time of termination, I have already planned a future research project involving the persons involved (Doris Arnold)? Or do I continue the established relationships because of a personal bond (“friends”/Navina Sarma) has developed? What responsibility do I carry as project initiator or researcher? Was I able to initiate positive developments and avoid the emergence of new dependencies by creating a common space of exchange and support?

It is clear from the PaSuMi and “Back to Life” projects that new networks for mutual support had formed among the persons involved—independently of the academic coordinators. It remains to be seen to what extent these new networks will become permanent in the long term or to what extent research participants may feel abandoned if there is no further regular contact after completion.

From the point of view of the “workshops” and the DIPEx project, on the other hand, it is important to bear in mind that participants in these research projects only wanted to be contacted with queries to a certain extent, as the persons in these projects were able and wanted to make an individually limited commitment for their participation. On the one hand, concerning the design of the research relationship, the question arises here of the individual “dosage” of involvement or participation or of closeness and distance in this relationship. Participants and researchers have different perceptions and needs in this regard. These needs must be made transparent and must be reflected upon individually in participatory research. This was done, for example, in the “Back to Life” project. However, unlike in conventional, positivist/post-positivist research, participatory researchers cannot retreat to their scientific distance.

On the other hand, it should be borne in mind that the research participants’ involvement in all four of the research projects presented required a different amount of time and effort. In the PaSuMi project, for example, community partners were also paid an expense allowance. Thus, in the participatory follow-up project to the “workshops”, it will be necessary to finance payment for the time spent on research activities as working time for the participating nurses in outpatient services or finance compensation for expenses for family caregivers (for persons to stand in for their care responsibilities) to enable their reliable participation in the research process. This is because these persons make a very similar and important contribution to data collection and analysis or to development processes as the researchers and can therefore also expect a payment in return that seems appropriate to them [27].

### 4.3. Understand Participation as a Two-Way Process That Requires Adjustments

An important potential of participatory research is to make voices audible or to make realities visible from the perspective of actors of groups that are often not heard clearly enough in public and scientific discourse [18]. In the DIPEx project, the researchers’ intention was to present the lifeworld and reality of persons affected by chronic illnesses from their own perspective and to give them the opportunity to have their say. However, the counter-reading of transcripts in the form of “Member Checks” was sometimes perceived by the participants as excessive and the reading of these texts, which arise from spoken language and cannot be compared with texts that are grammatically well formulated word for word in writing [62], caused discomfort for some respondents. Even the presented explanation and preparation for the unusual format of transcripts could only absorb this impression to a limited extent. An intensive examination of one’s own illness experience, sometimes written over 30 to 60 pages, can be emotionally and cognitively demanding, or even overwhelming for respondents [63]. From a research ethics perspective, there is also the question of the researcher’s responsibility to critically examine the procedure recommended in the methods manual for reasonableness [64]. Thus, the fact that those involved in the research did not want their voices heard and read in the way suggested to them became a problem in the DIPEx project, contrary to the intentions of the researchers.

The researchers in the DIPEx project would also like to avoid irritations of this kind among the participants in the future, so that the moment of the interview, which was experienced as positive, is not overshadowed negatively afterwards. Therefore, alternative approaches are currently being investigated in a participatory project with DIPEx participants. In this context, an “experienced-based co-design” [54] was used, which provided for participatory action and would give the participants options to determine the personally feasible level of participation. A possible suggestion for this could be to discuss selected text passages from the transcripts with the participants, which could be considered for publication on the website.

In contrast, the co-researchers in the project “Back into Life” presented their view of their reality by not only taking and discussing the photos, but also as independent actors and thus participated actively in the dissemination of the research findings. In the PaSuMi project, community partners took the space that was provided by the participatory approach to participate and shape the common project goal, but also to develop their personal skills and to manage their everyday life. Empowerment took place through the participants themselves and participatory research can help to recognize, activate, and use one’s own resources.

In the “workshops”, the participating actors contributed their respective expert knowledge to the research process. However, their involvement was limited to checking and, to a limited extend, to deciding whether their voices were heard and represented adequately. However, this did not enable them to participate in the research actively as co-researchers. The original plan for this project envisioned an approach more closely aligned with participatory research, which ultimately could not be implemented. More to the point, the initially intended involvement of participants in data analysis would ultimately have entailed the risk of mere tokenistic participation [9]. Because of the very limited time available, the participation would have had more the character of “assistance with research activities” without, however, opening sufficient possibilities for shared decision making in the research process, which was not possible under these circumstances.

## 5. Conclusions

The four research projects presented in this article provide a critical examination of the construction of relationships and of power dynamics in research processes that facilitated or limited possibilities for participation. These opportunities rested on different preconditions and resulted in different evaluations and experiences of researchers.

Relationships may take very different forms but remain the essential element of participatory research processes, where those persons who are at the center of research must have the opportunity to contribute their own experiences and perspectives to the research process on an equal basis. Thus, the distinctive features of participatory approaches are not only the democratic design of the research process itself but also their contribution for empowering the persons involved to overcome discriminating living conditions or hierarchically structured power dynamics in their lives.

The prerequisite for this is that persons representing science and practice see themselves equally as learners in the research process to be able to share their questions and uncertainties as well as to engage in joint reflection. Different expectations on both sides, different levels, and kinds of knowledge as well as issues of power or individual interests and goals can thus be taken up and dealt with.

In this article, scientists from different health professions acting in different research contexts shared, discussed, and reflected on their respective experiences with participative research to provide opportunities for other researchers to reflect on their own past and future research.

The practical implication of the article lies above all in the recognition that the critical self-reflection of participatory research processes prevents misinterpretations, provides deeper insights, and facilitates processes of change for everyone involved.

## Data Availability

Not applicable.

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
