# Peer review of "“What Do You Need? What Are You Experiencing?” Relationship Building and Power Dynamics in Participatory Research Projects: Critical Self-Reflections of Researchers"

_ijerph, 2022, doi:10.3390/ijerph19159336_

Round 1

Reviewer 1 Report

The article is of excellent quality and the topic is new and innovative. I leave some comments with adjustments to improve the reading and quality of the article:

- Present an international overview of the topic in the introduction (what has been done in a similar way in other countries, or even what has not been done);

- Improve the practical implications of the study at the end of the text (How important is the study for society? How important is the study for professionals? How important is the study for the health system? Etc..)

- The article is too long, I suggest giving a summary in some parts.

- I suggest removing references from the conclusion, leaving only the answer of your objective of the study. The conclusion cannot be referenced [reference 5]

- I suggest shortening the conclusion, it's too long. The conclusion only needs to be the answer to the purpose of your study.

Author Response

Author’s reply to Review Report (Review 1)            Manuscript - 1817229

Dear reviewer,

thank you very much for your kind appreciation of our article and for your most valuable advice for further improvement!

We revised the manuscript according to your recommendations as reported below.

Please note: Due to the recommended international overview of the topic (recommendation #1) and the recommended language editing, we decided to use a different English wording at several points in the text, including minor changes in the abstract (see response to recommendation #1).

We attached the revised and edited version of the manuscript for you for reference purposes (saved as "author coverletter").

The language improvement also prompted a suggestion of a change of the wording in the title of the article as follows:

  • line 1-4 (submitted version)
    “What do you need? What are you experiencing?" Relationship building and power relations in participatory research projects: critical self-reflections of researchers
  • line 1-4 (revised version)
    "What do you need? What are you experiencing?" Relationship building and power dynamics in participatory research projects: critical self-reflections of researchers

We hope that you approve of these changes. (See also response to recommendation #1).

Below we report the text referring to the revised version of the manuscript before we submitted it to the language editing. Since the editor only suggested few changes to the revised text, we did not transfer the changes to the revised text segments after editing and hope that this will not be a problem.

We hope that we were able to follow your recommendation with the changes made and the explanations in the point-by-point response to your satisfaction. If, however, there are points that are not covered or if questions arise, please do not hesitate to draw our attention to it.

Yours sincerely,

on behalf of all authors,

Doris Arnold

recommendation

response

1.       Present an international overview of the topic in the introduction (what has been done in a similar way in other countries, or even what has not been done)

We undertook a literature search for articles on research relationships or power relations in participative research on health issues and in the health professions (PAR, AR and PHR mostly). This is included into a partly rewriting of the introduction, while deleting some references to the German discourse. We referred to some articles in parts of the discussion as well.

(Line-numbers refer to the submitted version of the manuscript.)

·         line 77: (new text)
On the other hand, GREEN & JOHNS [9] warn against "token" participation and point to recent pressure from ministries to make projects participatory.

·         Line 78 - 89: (revised and supplemented text – original text deleted)
Looking at the issues compiled by ALTRICHTER and GSTETTNER on alleged and actual weaknesses of the action research approach today (in 2022), we find that many are still relevant and unanswered – in Germany and worldwide. This article will focus on the issue of "relations between researchers and researched" [6,10] (p. 69) raised by ALTRICHTER and GSTETTNER. In the English discourse on participatory health re-search, challenges connected with power and positionality of researchers in working with co-researchers are addressed in different contexts. Drawing on data from workshops with an international sample of participatory health researchers, EGID et al. [11] shed light on these issues and the need for reflexivity as a means for researchers to ad-dress power inequities in participatory research practice. SMITH et al. [12] reflect on different positionalities of class, gender, race, sexual orientation and the status of in-sider or outsider of communities, using episodes taken from the experiences of novice participatory action researchers. In the context of CBPR, MUHAMMAD et al. [13] dis-cuss issues of identity, intersectional positionality and power dynamics by means of autoethnographic self- reflections of experienced researchers in working with partners from marginalized communities. In a slightly different vein, JAGOSH et al. [14] show the crucial importance of trust, power-sharing and co-governance for CBPR projects to be successful, employing a qualitative analysis of interviews with community members and researchers.

In this article, we discuss individual experiences in four research projects in the German context. We look at relationships and power dynamics between persons engaged in research processes. By reflecting challenges and by showing how we understood and reacted to them, on one hand, special features and the high value of participatory health research are emphasized. On the other hand, limitations to these endeavors are made transparent.

We exchanged the translation of the German Version of the definition of PHR with the original English definition of the ICPHR.

·         line 97-102 (revised version – original text deleted):
The international Collaboration for Participatory Health Research (ICPHR) states the goal of PHR and the demands on participation in the research relationship within PHR as follows:

"The goal of PHR is to maximize the participation of those whose life or work is the subject of the research in all stages of the research process, including the formulation of the research question and goal, the development of a research design, the selection of appropriate methods for data collection and analysis, the implementation of the research, the interpretation of the results, and the dissemination of the findings. (…)” [17] (p. 6).

Furthermore, we edited the text using more appropriate wording, taken from relevant English articles, such as “power dynamics” instead of “power relations” or “power structures” in the submitted version.

Most importantly, we changed the wording of the title, changing “power relations” to “power dynamics”:

·         line 1-4 (submitted version)

"What do you need? What are you experiencing?" Relationship building and power relations in participatory research projects: critical self-reflections of researchers

·         line 1-4 (revised version)

"What do you need? What are you experiencing?" Relationship building and power dynamics in participatory research projects: critical self-reflections of researchers

We also applied some changes in the wording of the abstract to approve readability:

·         line 18-32 (submitted version, changes bold)

Participatory approaches create opportunities for cooperation, building relationships, gaining knowledge, and rethinking and eventually changing power structures. From an international perspective, the paper article first looks at the historical development of different participatory approaches in which . B building relationships and managing the balance of power between researchers and participants persons engaged in participatory research are central to participatory research. The authors present critically reflect on four research projects to show how they understood and implemented participatory research in different ways and what they have learned from their respective experiences. PaSuMi is a project in the context of addiction prevention with migrants and provides a glimpse into the different contexts of participatory research. The initiator of Back into life –with a power wheelchair with post-stroke individuals who use the assistive device in community mobility reflects on the shifting and intertwining of her multiple roles. In Workshops for implementation of expanded community nursing, expanded new professional roles for nurses in com-munity nursing were developed; here limitations to participation and ways to deal with them are illustrated. DIPEx deals with the challenges of enabling participation of persons with multiple sclerosis via narrative interviews on the experience of health and illness. All examples underline the necessity of a permanent reflection on relationships and power relations dynamics in participatory research processes.

2.       Improve the practical implications of the study at the end of the text (How important is the study for society? How important is the study for professionals? How important is the study for the health system? Etc..)

Our article comprises a critical methodological reflection on four different studies and, therefore, it differs somewhat from a report on a single empirical study. However, the findings of these reflections offer practical implications that are important for researchers engaging in participative research practice. Thank you for the suggestion of pointing out these practical implications from our reflections more explicitly.

We did so in pointing out lessons learnt by using additional subtitles in the discussion (chapter 4):

·      Between line 1068/1069 - 4.1 Building trusting relationships in participatory research processes

·      Between line 1183/1184 - 4.2 Participation in the area of tension between time and dosage

·      Between line 1219/1220 - 4.3 Understand participation as a two-way process that requires adjustments

Furthermore, we included lessons learnt in the conclusion (see below, recommendation #5)

3.       The article is too long, I suggest giving a summary in some parts.

We shortened the article as far as possible:

·         We shortened the text from a total of 18,424 words in the submitted version of the manuscript to a total of 16,676 words in the edited and revised version.

4.       I suggest removing references from the conclusion, leaving only the answer of your objective of the study. The conclusion cannot be referenced [reference 5]

We removed references from the conclusion (see below, recommendation #5)

5.       I suggest shortening the conclusion, it's too long. The conclusion only needs to be the answer to the purpose.

We rewrote and shortened the conclusions, focussing on our main practical implications or lessons learnt as recommended above (see recommendation #2).

·         line 1249 -1340 deleted, new text:

The four research projects presented in this article provide a critical examination of the construction of relationships and of power dynamics in research processes that facilitated or limited possibilities for participation. These opportunities rested on different preconditions and resulted in different evaluations and experiences of researchers.

Relationships may take very different forms, but remain the essential elements of participatory research processes, where those persons who are at the centre of research must have the opportunity to contribute their own experiences and perspectives to the research process on an equal basis. Thus, the distinctive features of participatory approaches are not only the democratic design of the research process itself but also their contribution for empowering the persons involved to overcome discriminating living conditions or hierarchically structured power dynamics in their lives.

The prerequisite for this is that persons representing science and practice see themselves equally as learners in the research process in order to be able to share their questions and uncertainties as well as to engage in joint reflection. Different expectations on both sides, different levels and kinds of knowledge as well as issues of power or individual interests and goals can thus be taken up and dealt with.

In this article, scientists from different health professions acting in different re-search contexts shared, discussed and reflected on their respective experiences with participative research in order to provide opportunities for other researchers to reflect on their own past and future research.

The practical implication of the article lies above all in the recognition that the critical self-reflection of participatory research processes prevents misinterpretations, provides deeper insights and facilitates processes of change for everyone involved.

Reviewer 2 Report

Addressing the limits and constraints of participatory research as well as its appropriateness in some themes-collectives is stimulating. To do it from the comparison of how the referred problems have been managed from the experience of execution in four projects seems to me to be correct. However, I must say that the format, as it is presented, is more appropriate for a book chapter than for an article. The text is excessively long and it would be recommended to reduce it. I am aware of the authors' need to contextualize the projects in order to understand their scope and limitations, especially the methodological ones on which the article reflects. Even so, there are parts of the discussion that are repeated and could be deleted in the presentation of the projects. 

As for the methodology of analysis-discussion, it would be advisable to elaborate a section reflecting the procedure to be followed by the authors (criteria or central elements of the discussion, which are deduced, limitations and proposals for overcoming these limitations, addressed in some cases but not in all).

Author Response

Author’s reply to Review Report (Review 2)                              Manuscript - 1817229               

Dear reviewer,

thank you very much for your kind appreciation of our article and for your most valuable advice for further improvement!

We revised the manuscript according to your recommendations as reported below.

Please note: Due to the inclusion of more international literature on the topic and the language editing recommended by another reviewer, we decided to use a different English wording at several points in the text, including minor changes in the abstract.

We attached the revised and edited version of the manuscript for you for reference purposes (saved as "author coverletter").

The language improvement also prompted a suggestion of a change of the wording in the title of the article as follows:

  • line 1-4 (submitted version)
    “What do you need? What are you experiencing?" Relationship building and power relations in participatory research projects: critical self-reflections of researchers
  • line 1-4 (revised version)
    "What do you need? What are you experiencing?" Relationship building and power dynamics in participatory research projects: critical self-reflections of researchers

We hope that you approve of these changes.  

Below we report changes in the text referring to the revised version of the manuscript before we submitted it to the language editing. Since the editor only suggested few changes to the text, we did not transfer the changes to the revised text segments after editing and hope that this will not be a problem.

We hope that we were able to follow your recommendation with the changes made and the explanations in the point-by-point response to your satisfaction. If, however, there are points that are not covered or if questions arise, please do not hesitate to draw our attention to it.

Yours sincerely,

on behalf of all authors,

Doris Arnold

recommendation

response

1.       The text is excessively long and it would be recommended to reduce it. I am aware of the authors' need to contextualize the projects in order to understand their scope and limitations, especially the methodological ones on which the article reflects.

Even so, there are parts of the discussion that are repeated and could be deleted in the presentation of the projects.

We removed parts of the description of the four projects that are also mentioned in the discussion. Furthermore we rewrote and shortened the conclusion (see recommendation #2). Overall, we shortened the text from a total of 18,424 words in the submitted version of the manuscript to a total of 16,676 words in the edited and revised version.

Line-numbers refer to the submitted version of the manuscript.

Reductions of text redundancies and minor modifications in the PaSuMi part (Chapter 3.1, Line 223-398)

·      Line 234-235 text deleted (submitted version of manuscript): PaSuMi was the first participatory project I coordinated as a researcher.

·      Line 255-256 part of the text deleted: The Canadian HIV/AIDS Legal Network for instance states, that persons who use drugs know best what works in their community whereas others might not know much about it.

·      Line 261-264 text deleted: An entry from my research diary illustrates this: "Now I see how difficult my role is: I have to give constructive criticism, somehow lead and accompany the project and should not question the expertise of the project partners. After all, they are experts in their field" (entry from my research diary on 25.08.2017).

·      Line 319-321 part of text deleted: Similar experiences were also reported from the local projects. For example, in some project teams, the research community partners decided that all meetings and trainings should be in German because they wanted to use the meetings to learn German. They requested various trainings because they wanted to learn how to give presentations and front events with confidence.

·      Line 339-342 text deleted: All this is an expression of a kind of relationship that does not exist in other forms of research. Community partners took the space that was provided by the participatory approach to participate and shape the common project goal, but also to develop their personal competencies and to manage practical life tasks.

·      Line 394-350 text deleted: Finding a good balance here was a huge challenge.

·      Line 361-368 text deleted: On the one hand, this found expression in the cross-site joint PaSuMi finalmeeting at the end of 2019, in which the self-organized initiative BerLUN awarded honorary certificates to individuals from other local PaSuMi teams -one went to two community partners whose deportation could not be prevented in the course of the projecteven by the PaSuMi team, and one to a peer who had become clean in the course of the project and who made a decisive contribution in advancing the project in his institution. Secondly, a meeting of the community partners of the PaSuMi project took place in 2020, almost one year after the end of the project, and another one in 2021.

Reductions of text redundancies and minor modifications in the “Back into Life” example (Chapter 3.2, Line 396-625)

·         Line 407 : Many persons stroke survivors experience permanent limitations in their activities of daily living, due to acquired brain injury including mobility

·         Line …. presented them to each other in the meetings, discussed and evaluated analyzed them together

·         Line 435: In the project, former rehabilitation participants were included as co-researchers. All of them had co-researcher had ...

·         Line 480: At the time of invitation or cooperation in the research project, a therapeutic relationship with the former rehabilitants no longer existed.

·         Line 572: Another goal and learning process at the same time was the sharing of responsibility for the joint research process and communicating this all the time.

·         Line 585: In the course During of the entire research process, it became clear that there was space for both roles

·         Line 605: The collaboration between the author and the co-researchers took place in five group meetings, the joint meetings in their respective living environment or "data collection" (recording their concerns via photographs) and the joint utilization of the results in the form of exhibitions and lectures.

·         Line 615: Due to a case of illness of a co-researcher, a planned joint project presentation had to be changed.

Reduction of text in the “Workshops” example (Chapter 3.4 (revised version), 3.3 (submitted version)

·         line 658-661 deleted:
This scientific continuing education program and the subsequent efforts to implement extended nursing practice performed by its graduates want to contribute to the professionalization of Nursing, especially for the community care sector, which has been rather marginalized in German nursing so far.

·         line 664-666 deleted:
Moreover, the provision of such care services is of major importance for coping with the enormous future care needs in the community sector in the context of the current and future demographic changes in Germany, Europe and worldwide.

·         line 682-686 deleted:
The challenge is therefore that extended nursing care for persons with dementia cannot be implemented in the practice of community care under the prevailing conditions in the health care system, although there is a demonstrable need for this and caregivers can be appropriately qualified for these activities, e.g., as part of the aforementioned certificate program.

·         line 710-725 deleted
The aim of the pilot project was to find feasible solutions so that nurses who had attended the certificate program could subsequently apply the knowledge they had acquired there for the provision of high-quality community nursing care for persons with dementia in their everyday work. It was assumed that this expanded community nursing practice based on a scientific qualification would firstly lead to greater satisfaction for nurses in their work, and secondly provide greater recognition for them as members of a so-called women's profession, especially since nurses working in the community belong to the marginalized groups within the nursing profession.

A nurse who works as a nurse manager in a community nursing service put it this way:

"It would be an idea to accompany a family, not just when everything’s agog, but so that trust can be built up beforehand or ... persons get to know each other. I also think it is a good idea for us nurses, I'd really like to do that, rather than administering compression stockings and medication by the clock. I can imagine that well, to accompany a family, I could imagine many employees who see more meaning in the work through that." [53] (p. 201, translation by the authors)

·         line 726-727: Chapter (3.3.3/3.4.3)  merged with chapter (3.3.4); title changed accordingly;
new title: 3.4.3 The pilot project: Opportunities for participation and barriers to participation in the research process
deleted:  Working on ideas for implementing expanded community nursing practice

·         line 746 title of chapter (3.3.4) deleted:
Opportunities for participation and barriers to participation in the research process

·         line 731-734 : deleted
The first two workshops focused on the development of expanded nursing tasks and professional roles in community nursing for the graduates of the certificate program, while the third workshop discussed possibilities for financing these tasks.

·         line 804-810 deleted:
I see our task as researchers first and foremost in establishing a sustainable relationship with these nurses and in working with them as equal partners in the research process, and ultimately involving them in joint publications. Other persons participating actively as co-researchers should be persons with dementia themselves and their caregivers. In addition, managers and staff in the teams of community nursing services, family physicians and other relevant stakeholders should be involved in the research process.

·         line 822-824 deleted:
The implications of a participatory research project in this institutional context should also be considered in relation to the reflection of power structures and research relations in comparison with other participatory research projects presented in this paper.

Reductions of text redundancies in the DIPEx example (Chapter 3.5 (revised version), 3.4 (submitted version)

·      Line 846 – 849 text deleted
In the following, the research project "DIPEx.ch" is first presented in relation to its structure, objectives and partly the processing of the results. Subsequently, experiences from the perspective of the researchers with participatory elements within the research process are described.

·      Line 921 – 924 sentence shortened

received Swiss-wide approval from the cantonal ethics committee with the number BASEC Req-Nr. 2018-00050.

·      Line 926 – 927 text deleted

and are analyzed according to BRAUN and CLARKE [59,60] until data saturation
59 Braun, V.; Clarke, V. Successful qualitative research: A practical guide for beginners; Sage: London, UK, 2013.
60.                Braun, V.; Clarke, V. Using thematic analysis in psychology. Qualitative research in psychology 2006, 3, 77–101.

·      Line 939 – 941 text deleted
With reference to the participatory element and regarding the requirements for implementation, the project shows parallels to the described project 2 "Back to life - with the power wheelchair" in this paper.

·      Line 945 – 947 text deleted
The transcripts were slightly smoothed linguistically or adapted to High German according to the rules but still reflected the everyday language.

·      Line 953 – 954 text deleted

The interview is not an everyday situation, but a moment on its own in a certain mood and temporally self-contained.

·      Line 957 – 959 text deleted and modified as follows:

Reading the transcript, partly over many pages, also confronts one with oneself, one's own history in a context that is not commonplace. Reading one‘s own transcript is not only about

·      Line 971 text deleted and modified as follows:

Experiences show that reading it triggers further discomfort in the argument. This is also As shown by LUCIUS-HOENE….

·      Line 978 – 980 text deleted and modified as follows:

It requires an examination of one's own research contribution and a balanced relationship of closeness and distance between the researcher and the participants.

Within this relationship, the question of the individual "dosage" or closeness and distance in this relationship also arises. Participants and researchers have different perceptions and needs in this regard. Examples of designing this relationship are that contact is maintained by e-mail or telephone afterwards on the course of the condition, as well as the reference back to the text design of the transcript or in the further course of the project for the elaboration of the texts for the website. This shows that p

·      Line 1018 – 1024 text deleted:

In part, the researchers' expectations of the participatory procedure were met with rejection by the participants, because they felt it was overtaxing in the process. The willingness to be interviewed about their own experience was great and was affirmed. Being further involved and actively shaping the process was not of concern for or the idea of all participants. The aspect that one's own patient narrative could be of interest to others was not a priority for all participants.

Reductions of text redundancies and minor modifications in the Discussion (Chapter 4, Line 1053-1266)

·         Line 1054-1068 text partly deleted and modified as follows:

With this contribution, we intend to reflect critically on our own project work and findings. We considered how relationships in participatory research processes can be discussed in terms of power relations between participants, methodology and procedures, initiation of projects, mutual learning, political goals, or different interests [5].

Closely related to the aspect of building relationships in participatory research projects is the importance of participation as taking an active part in research. In this regard, interesting differences were described in each of the four contributions. In these projects enabling persons who were at the center of the research processes to present their reality from their own perspective in a self-determined way succeeded in different ways and with varying degrees of success. Based on the four examples presented our central concern is to identify, name and reflect on the numerous factors that influenced the course of the concrete projects in their respective contexts in order to further the development of participatory research. In the following, we discuss these factors in relation to the shaping of relationships and the level of participation achieved, and highlight both the various success factors and the challenges in participatory research processes.

·         Line 1052: The experiences and reflections from the four different projects described here show that the establishment of trusting relationships between scientists academics as researchers and other research participants is a central element and a foundation for the success of a participatory project, as is shown in other research about CBPR [14].

·         Line 1055 text party deleted as follows:

At the same time, it is evident that this process has taken up different amounts of space, time and weight in the four projects and that it could build on different preconditions.

·         Line 1064 text party deleted as follows:

Thus, various migrant individuals and groups and persons with severe acquired brain injury after stroke could be successfully involved, while in the "Workshops" the lack of existing relationships and networking also limited access to nurses in community nursing services and thus their recruitment for participation in the research project.

·         Line 1090-1102 text partly deleted and modified as follows:

Similarly, Ddespite partly pre-existing relationships in PaSuMi and "Back into Life", the new kind of cooperation as partners desired on an equal footing had to be negotiated to overcome problems with positionality in previously existing hierarchically structured power relationships in the respective work contexts. This was not achieved through one joint meeting but grew gradually during the project, primarily through sufficient sensitivity, knowledge, and openness to the different realities of life and experiences made, as well as in encounters outside the joint working meetings. Both authors describe how opportunities were created for activities desired by the research participants outside the originally planned joint working meetings repeatedly. This ranged from visiting an exhibition together or dining in a restaurant to direct support in writing job applications or applying for a new certificate with a token for free use of public transportation. Thus, in these projects, also concerns of the persons involved were repeatedly implemented, which at first glance had nothing to do with the goal of the research project were considered. According to Navina Sarma and Tabea Böttger, formulating and dealing with (personal) needs of research partners was part of the participatory process and supported trust building.

·         Line 1116-1134 text deleted:

The examples of the PaSuMi and "Back to Life" projects show both similarities and differences between participatory research and ethnography. In ethnographic research the establishment of "access to the field," that is, the building of personal relationships with participants, as well as the intensive reflection on the role of the ethnographer and these relationships are equally important for the success of the project and often provide valuable insights into the research question itself [11,61–64]. The two research styles differ, among other issues, in the design of the power relationship between participants and ethnographic researchers on the one hand, and between researchers and other research participants in participatory research projects (e.g., experts in their own right; see [8]) on the other. In ethnographic research, building a trusting relationship with participants is important as well, but participation in the research process in ethnographic studies is usually limited to having a say. Thus, participants in ethnographic studies may well be involved in intensive discussion of data from participant observation (see, e.g., [11], p.181f), but there is no claim to enable research participants to participate regularly in data collection and analysis or to have an equal say in decisions in the research process, as is characteristic of participatory research. Ultimately, the goal of ethnographic research is to gain knowledge about the social worlds in which the participants live and work, while participatory research also works toward initiating processes of change within the living or working conditions of the research participants.

·         Line 1135-1137 text partly deleted as follows:

The PaSuMi and "Back to Life" projects make it clear that, in addition to pre-existing relationships, the number of joint working meetings over a longer period (12 and 24 months, respectively) contributed significantly to relationship building.

·         Line 1155-1158 text deleted:

"I understand expert knowledge from lived experience as equivalent to the professional expertise of practitioners and the methodological expertise of scientists", as Navina Sarma summarizes it in her contribution.

·         Line 1166-1170 text deleted:

The differentiated look at what is possible and desired for the individual person involved in participatory research prevents egalitarianism and enables "different scopes for co-design (...), which are located in the area of tension between enabling influence and avoiding excessive demands" (Navina Sarma in her contribution).

·         Line 1247-125337 text partly deleted and modified as follows:

In contrast, the co-researchers in the project "Back into Life" not only presented their view of their reality by not only taking and discussing the photos, but also presented it as independent actors and thus participated actively in the dissemination of the research findings. In the PaSuMi project, community partners took the space that was provided by the participatory approach to participate and shape the common project goal, but also to develop their personal skills and to manage their everyday life the community partners themselves also presented their view of their realities. They took over decision power in the research process and over the findings and ensured that they could use the project to acquire, among other things, appropriate knowledge of the German language and skills for presentations.

2.       As for the methodology of analysis-discussion, it would be advisable to elaborate a section reflecting the procedure to be followed by the authors (criteria or central elements of the discussion, which are deduced, limitations and proposals for overcoming these limitations, addressed in some cases but not in all).

We added a description of the approach adopted by the authors when engaging in the critical reflection of their respective projects, addressing different criteria and central elements of these reflective processes. This is placed at the beginning of chapter 3, before entering into the presentation and reflection of the individual examples.

·         Line 189-222 (deleted)

Based on four examples from our research, we will critically discuss participation in research practice and the shaping of relationships within the research process. All projects are independent of each other in terms of time, content and aims. The projects presented here establish a direct connection to lived research practice. They highlight potentials as well as challenges of participatory research. This reflects the discussions and reflections on participation and participatory research that took place during an ICPHR Training in Participatory Health Research program in 2018 at the Catholic University of Applied Sciences, Berlin.

Project example a) PaSuMi (participation, addiction prevention and migration), a nationwide participatory model project in the context of addiction prevention with migrants, provides insight into the reflection process of the coordinating researcher. The focus is on how to deal with relationships that have developed, the question of re-linquishing power in participatory processes, and the dilemma of one's own role and expectations.

Project example b) entitled "Back into life - with a power wheelchair" vividly il-lustrates the role played by the existence of a trusting relationship as the basis for par-ticipation in this research project. Without the pre-existing relationship between the research initiator and the research participants, the project would not have come about in this form in the view of the author.

Project example c) on "workshops for the development of extended professional roles in community nursing" describes experiences from a research project with the aim of implementing extended professional roles for nurses in community nursing. On one hand, it became apparent that the facilitation of co-operation of participants in participatory research projects rests on certain preconditions. On the other hand, the results of the ultimately realized pilot project with involvement of stakeholders are very important for the chances of success of a future participatory implementation project, in which co-operation of nurses as co-researchers implementing their own ex-tended professional roles should be facilitated.

Project example d) "DIPEx" (Database of Individual Patients' Experiences), the fourth contribution, shows research experiences from an ongoing international project reported from Switzerland. The project focuses on experiences and perspectives on the experience of health and illness through narrative interviews. Thereby, excerpts of the project method are reflected from the perspective of the researchers on the shared process experience and the relationship design.

·         Line 189ff new text:
3.1 Approach to critical self-reflection on research experiences 

The following examples are based on discussions and critical reflections on participation and participatory research that took place during an ICPHR Training in Participatory Health Research program in 2018 at the Catholic University of Applied Sciences, Berlin. We present four of the projects that were accompanied by all participants during the training in the form of collegial consultation. All projects are independent of each other in terms of time, content and aims. We chose an approach of critical self-reflection that in a similar way has been used by other researchers for sharing experiences and enhancing knowledge about the practice of participatory research [12,13]. First, we shortly introduce the aims, design and context of each project and describe their role as researchers or initiators of the projects. In a second step, we critically reflect our personal experiences as well as challenges during the research process that relate to issues of power dynamics and relationships. Finally, we point out aspects that facilitated or limited participation in research processes.

We discussed the approach adopted by the authors when engaging in the critical reflection in the Discussion (Chapter 4, Line 1053-1266) as follows:

·         We added the following new text after line 1089:

This underlines limitations posed by deadlines of funded projects that cause too little time and resources to facilitate more than tokenistic participation in applied health re-search mentioned by GREEN and JOHNS [9].

In the context of CBPR, the need to work and reflect on the positionality of re-searchers when building trust with members of marginalized communities has been shown [13,21]

We added subheadings to the discussion to increase readability:

·      Between line 1068/1069 - 4.1 Building trusting relationships in participatory research processes

·      Between line 1183/1184 - 4.2 Participation in the area of tension between time and dosage

·      Between line 1219/1220 - 4.3 Understand participation as a two-way process that requires adjustments

Furthermore, we rewrote and shortened the conclusion focussing more on implications of our reflections that aim at lessons learnt for the success of participatory research or reacting to some of the limitations we found.

·         line 1347 -1396 deleted

The four participatory research projects presented in this article had different preconditions and resulted in different evaluations and experiences of researchers show a critical examination of the possibilities of building relationships. Based on the observations and the still helpful criticism of ALTRICHTER AND GSTETTNER [5], it becomes clear from the examples discussed and reflected on here that mutual learning certainly had taken place, but that roles and power relations also had been critically questioned. In this way, a deprivation of the right of decision-making of persons involved in research [5] was largely prevented from the outset.

Overall, it can be stated that the building of relationships is a supporting, maybe even the most supporting element of participatory research processes. However, the possibilities do not only depend on the project context or the participants' under-standing of participation. Above all, it is also important which possibilities are given to the persons who are at the center of the research to contribute their perspectives on "reality" as self-determined and as much on an equal footing as possible. These opportunities are closely linked to the insight that the potential of participatory approaches lies not only in the democratic design of the research process, but above all in the chance to make a direct contribution to overcoming hierarchically structured power relations. It is only through the joint attempts to establish equality between researchers and “researched” that quite different forms of power can be recognized, questioned and perhaps even overcome in parts.

Relationships – regardless of whether they already existed at the beginning of the research or whether they have to be established first - have to be redesigned and changed in the course of the process repeatedly, since roles can and should change (among other things with a view to the desired empowerment of all persons involved in the research). This raises the question of closeness and distance throughout the re-search process. However, this should always be decided individually and according to the possibilities and capacities of the participants and the conditions of the project. Relationships that already existed before the start of a joint research process – even if they have to be redesigned and, above all, constantly reflected upon – nevertheless of far-reaching possibilities, for example with regard to individual accessibility or the accessibility of networks.

The choice of method is rather less important in the context of relationships in re-search. However, it seems important to examine the extent to which individual methods could overtax participants or whether methods other than those planned would be preferred, jointly.

If both science and practice (i.e., the groups and individuals involved in the re-search process) strived to define themselves primarily as learners and acted accordingly, an essential precondition for the establishment of equality in research would ensue. In the foreground, however, is the research process that is co-designed, co-determined, and reflected upon by all participants on an equal footing, which distinguishes participatory research from similar approaches (for example, ethnographic research). The reluctance and insecurity of all participants in participatory research processes, which can be seen and felt again and again in the examples, is not only due to possibly too high expectations from both sides (i.e., practice and science). In this context, the question also arises to what extent all participants are actually at a similar level of knowledge, who has what kind of power, and whether common goals are (could be) pursued in addition to individual goals. In shaping the inevitably different roles (while striving for equality), it therefore seems less important who is involved in what and to what extent, but rather the degree of trust that could be achieved as an important element of relationships. Researchers responsible for participatory projects need to ensure that those factors that hinder or promote relationship building, as shown in the examples, are seen and addressed in the research process. Time must be scheduled for this in advance, and disruptions must be perceived and taken seriously.

·         line 1249ff: (new text)
The four research projects presented in this article provide a critical examination of the construction of relationships and of power dynamics in research processes that facilitated or limited possibilities for participation. These opportunities rested on different preconditions and resulted in different evaluations and experiences of researchers.

Relationships may take very different forms, but remain the essential elements of participatory research processes, where those persons who are at the centre of research must have the opportunity to contribute their own experiences and perspectives to the research process on an equal basis. Thus, the distinctive features of participatory approaches are not only the democratic design of the research process itself but also their contribution for empowering the persons involved to overcome discriminating living conditions or hierarchically structured power dynamics in their lives.

The prerequisite for this is that persons representing science and practice see themselves equally as learners in the research process in order to be able to share their questions and uncertainties as well as to engage in joint reflection. Different expectations on both sides, different levels and kinds of knowledge as well as issues of power or individual interests and goals can thus be taken up and dealt with.

In this article, scientists from different health professions acting in different re-search contexts shared, discussed and reflected on their respective experiences with participative research in order to provide opportunities for other researchers to reflect on their own past and future research.

The practical implication of the article lies above all in the recognition that the critical self-reflection of participatory research processes prevents misinterpretations, provides deeper insights and facilitates processes of change for everyone involved.
